# Insights into the Epigenetic Basis of Plant Salt Tolerance

**DOI:** 10.3390/ijms252111698

**Published:** 2024-10-31

**Authors:** Dongyu Zhang, Duoqian Zhang, Yaobin Zhang, Guanlin Li, Dehao Sun, Bo Zhou, Jingrui Li

**Affiliations:** 1College of Future Technology, China Agricultural University, Beijing 100193, China; dy_zhang@cau.edu.cn (D.Z.); duonix@cau.edu.cn (D.Z.); zybing@cau.edu.cn (Y.Z.); 2023333020320@cau.edu.cn (G.L.); swxyzz@cau.edu.cn (D.S.); 2State Key Laboratory of Plant Environmental Resilience, College of Biological Sciences, China Agricultural University, Beijing 100193, China

**Keywords:** salt stress, epigenetic regulation, histone modification, DNA methylation, histone variant, non-coding RNAs

## Abstract

The increasing salinity of agricultural lands highlights the urgent need to improve salt tolerance in crops, a critical factor for ensuring food security. Epigenetic mechanisms are pivotal in plant adaptation to salt stress. This review elucidates the complex roles of DNA methylation, histone modifications, histone variants, and non-coding RNAs in the fine-tuning of gene expression in response to salt stress. It emphasizes how heritable changes, which do not alter the DNA sequence but significantly impact plant phenotype, contribute to this adaptation. DNA methylation is notably prevalent under high-salinity conditions and is associated with changes in gene expression that enhance plant resilience to salt. Modifications in histones, including both methylation and acetylation, are directly linked to the regulation of salt-tolerance genes. The presence of histone variants, such as H2A.Z, is altered under salt stress, promoting plant adaptation to high-salinity environments. Additionally, non-coding RNAs, such as miRNAs and lncRNAs, contribute to the intricate gene regulatory network under salt stress. This review also underscores the importance of understanding these epigenetic changes in developing plant stress memory and enhancing stress tolerance.

## 1. Introduction

Soil salinization is a critical issue in agriculture, with research by Negacz et al. indicating that approximately 17 million square kilometers of soil are affected by salinity [1]. This underscores the need to enhance crop salt tolerance to ensure global food security. Under salt stress, plants can regulate gene expression through epigenetic regulations, which involve alterations in chromatin architecture without changes in DNA sequences [2]. The well-known epigenetic regulatory strategies include DNA methylation, histone modifications, histone variants, and non-coding RNAs, all essential for plant adaptation to high-salt environments. DNA methylation, which adds a methyl group to the DNA sequence, is vital for gene expression regulation and environmental stress responses, particularly in plants with complex genomes [3]. There are many types of DNA methylation. We primarily focus on 5mC in this review. Histones, as proteins that package DNA, are regulated by post-translational modifications, such as methylation and acetylation, and histone variants, including H2A.Z and H2A.X, to modulate chromatin structure and accessibility [4]. Non-coding RNA (ncRNA), such as long non-coding RNAs (lncRNAs) and microRNAs (miRNAs), do not encode proteins but contribute to the complex gene regulatory network under salt stress by modulating key genes involved in hormonal response pathways and stress signaling. These epigenetic regulations not only modulate gene expression and rapidly adjust the physiological responses, enhancing their salt tolerance, but also play a crucial role in establishing plant stress memory. This enables plants and their offspring to adapt more swiftly to recurring stress. In summary, epigenetic regulation is pivotal in the plant response to salt stress, affecting both immediate physiological reactions and long-term adaptability and evolution. This knowledge provides vital strategies for agricultural breeding, facilitating the development of salt-tolerant crop varieties and contributing to global food security.

## 2. DNA Methylation

### 2.1. DNA Methylation in Plants

There are various types of DNA methylation, with 6mA and 5mC being the most extensively studied [3]. Among these, 5mC is notably prevalent under salinity stress. The 5mC modification is commonly observed in all three contexts of plant DNA sequences: symmetrical CG and CHG, as well as asymmetrical CHH (where H represents A, T, or C). Distinct sequence contexts necessitate specific methylases for maintenance. For instance, MET1 (methyltransferase 1) is charged with responsible for maintaining CG methylation, while CMT2 (chromomethylase 2) and CMT3 (chromomethylase 3) are involved in maintaining methylation at CHG sites. CHH methylation is regulated by DRM2 or CMT2, depending on the genomic context [3,5]. DRM2 is known to catalyze CHH methylation at RdDM (RNA-directed DNA methylation) target regions [6], while CMT2 maintains CHH methylation at heterochromatin containing histone H1.

Studies indicate that alterations in DNA methylation are closely linked to promoters and transposable elements (TEs) [7,8]. DNA demethylation occurs through two main mechanisms: active demethylation and passive demethylation. The processes of these mechanisms differ between plants and animals [9]. In plants, passive demethylation occurs during DNA replication and is attributed to either reduced DNA methyltransferase activity or lacking methyl donors [10]. Active demethylation, on the other hand, involves the enzymatic removal of methylated cytosine, facilitated by DNA glycosylases such as Repressor of Silencing 1 (ROS1), Demeter (DME), Demeter-like 2 (DML2), and Demeter-like 3 (DML3) [11]. This process is subsequently completed through a basal excision repair (BER)-dependent mechanism [12]. Increasing evidence suggests that active DNA demethylation is crucial in various biological processes across plant species, including fruit ripening, auxin-mediated development, and responses to environmental stress [13].

*S*-Adenosyl-L-methionine (SAM) serves as a crucial methyl donor for cytosine and lysine methylation in plants [14]. The tomato synthase of *S*-Adenosyl-L-methionine is induced by salinity stress, which positively correlated with increased lignin deposition in the vascular tissues under such conditions [15]. Furthermore, overexpression of tomato SAM synthetase *SlSAMS1* enhances salt tolerance by promoting gene body DNA methylation and the expression of the circadian rhythm core component *SlGI* [16,17]. Similarly, the downregulation of barley SAM synthetase *HvSAMS3* significantly impairs plant tolerance to drought and salt stress [18]. These findings underscore the potential regulatory role of SAM in plants subjected to salt stress.

### 2.2. Global Alterations of DNA Methylation under Salt Stress

Salt stress has been demonstrated to globally impact DNA methylation across various plant species, leading to alterations in gene expression. In alfalfa (*Medicago sativa*), a genome-wide increase in DNA methylation was observed under salt stress, particularly under high-salinity conditions [19]. The high-salt environment may influence DNA methylation levels by affecting the activity of C5-methyltransferases (C5-MTases) or other DNA methyltransferases [20]. In *Pyrus betulaefolia* (a wild pear), methyltransferases (MTases) and salt-responsive genes were upregulated in response to a high-salt environment. The DNA methylation levels of salt-responsive genes were altered under salt stress, and the precise mechanisms by which methyltransferases regulate the salt response in *P. betulaefolia* remain unclear [21].

Salt stress induces changes in DNA methylation levels in plants, with salt-tolerant plants generally exhibiting increased methylation and salt-sensitive plants showing decreased methylation [22]. This suggests that DNA methylation is crucial for plant adaptation to salt stress. Recent studies indicate that hypermethylation occurs at a significantly higher rate in salt-tolerant accessions in comparison with salt-sensitive ones in sugar beet. Thus, hypomethylation is more commonly observed in salt-sensitive accessions of sugar beet [23,24]. In cotton, under salinity treatment, various types of DNA methylation were observed in three cultivars: salt-tolerant CCRI 35, Zhong 07, and salt-sensitive CCRI 12. Generally, the global DNA methylation levels of the salt-tolerant cultivars CCRI 35 and Zhong 07 significantly increased under salinity stress, while no significant change was noted in the salt-sensitive cultivar CCRI 12 [24]. Additionally, salt stress caused a notable decrease in DNA methylation in the salt-sensitive rice cultivar IR29, compared to the salt-tolerant cultivar FL478 [25]. These observations suggest that the salt-tolerance plant may possess a unique mechanism that enhances DNA methylation under high-salinity conditions. However, the specific mechanism by which salt stress affects DNA methylation remains unclear and represents a potential avenue for future research. Contrarily, in some cases, the level of 5mC methylation increased in the salt-sensitive wheat cultivar, whereas it decreased in the salt-tolerant wheat cultivar SR3 under salt stress [26].

Alterations of DNA methylation induced by salt stress exhibit both tissue and growth stage specificity. Differences in the DNA methylation of genes related to growth and development between two different rice varieties result in significant variations in their salt tolerance [25]. Konate et al. found that the frequency of DNA methylation increased in barley leaves compared to roots, indicating that salt-induced DNA methylation is tissue-specific [27]. Additionally, Chen et al. reported that under salt stress, 61.2% of CGs, 39.7% CHG, and 3.2% CHH sites Glycine roots were methylated, with these levels being significantly lower than those in control conditions [28].

### 2.3. Regulation of Key Stress-Responsive Genes by DNA Methylation

High-salinity environments induce genome-wide DNA hypermethylation of transposable elements (TEs), with cytosine methylation within TEs accounting for nearly one third of all cytosine methylation [29]. OsBAG4 is a key regulator associated with DNA methylation of TEs [30]. Previous research has demonstrated that OsBAG4 functions as a positive regulator of salt-stress tolerance by acting upstream of *OsHKT1;5*, which encodes the Na^+^ transporter essential for expelling Na^+^ from leaves and maintaining Na^+^/K^+^ homeostasis under salt stress [31]. OsMYB106, a MYB transcription factor, and OsSUVH7, a DNA methylation reader, both interact with OsBAG4, enhancing its DNA binding affinity and forming a stable complex. OsMYB106 binds to the cis-regulatory sequences of *OsHKT1;5*, while OsSUVH7 associates with the methylated miniature inverted-repeat transposable element (MITE) [32] (Figure 1). *HKTs* also play a crucial role in wheat. Kumar et al. have identified genotype- and tissue-specific increases in DNA methylation triggered by salt stress, leading to the downregulation of *TaHKT2;1* and *TaHKT2;3* expressions in both shoot and root tissues of the wheat cultivars Kharchia-65, therefore enhancing salt tolerance [33] (Figure 1). In Arabidopsis, heavy methylation of the *HKT1* promoter across different sequence contexts may inhibit transcription in leaves and roots, whereas non-CG methylation may contribute to the fine regulation of *HKT1* expression in leaves (Figure 1). This regulation is crucial for long-term adaptation to salt stress but is not essential for short-term salt tolerance [22]. Besides *HKTs,* other salt-responsive genes regulated by DNA methylation include the flavanol synthase genes *TaFLS1* and *TaWRS15* in wheat and barley (*Hordeum vulgare*) [26,34,35].

### 2.4. Plant Stress Memory by DNA Methylation

The MTases also play a crucial role in heritability, as they enable DNA methylation changes induced by environmental disturbances to be sustained over time or passed to subsequent generations. This allows both the original organism and its progeny to better adapt to recurring environmental conditions. The adaptive phenomenon, which enables plants to “remember” past experiences and recall mechanisms of responding to environmental stress, is referred to as plant stress memory (PSM) [36]. Using *Arabidopsis thaliana* as a model, approximately 75% of salinity stress-induced differential methylated cytosine positions are inherited, although some of these changes may be lost in future generations [37] (Table 1). In addition to salt stress, CHG demethylation observed in heavy metal-treated rice leaf tissues can be transmitted to the next generation, indicating a meiosis-based inheritance mechanism [38]. Similarly, heavy metal-induced methylation changes in TEs are also inherited in rice [39]. In the annual plant *Polygonum persicaria*, heritable DNA methylation changes have been observed as well. Longer root systems and greater biomass are exhibited in the progeny of drought-stressed parents in comparison with progeny from non-drought-treated parents of the same genetic line [26,40]. Moreover, drought-induced changes in DNA methylation are inherited in subsequent generations, which modulate the expression of drought-responsive genes [41,42]. Most memory studies have focused on 5mC modification, leaving the role of 6mA largely unexplored. Notably, the number of generations over which DNA-methylated memory can be inherited is also an interesting question. Further exploration is needed to elucidate the molecular mechanisms underlying the formation and maintenance of plant stress memory.

## 3. Histone Methylation

### 3.1. Histone Methylation in Plants

The stability of chromatin is enhanced through the interaction between the negatively charged phosphate groups of DNA molecules and the positively charged amino acids in histone proteins. Post-translational modifications (PTMs) of both histone tail regions and histone fold domains are essential to regulating chromatin structure and its accessibility for various biological processes [48]. Histone methylation influences the association between DNA and histones by altering local hydrophobicity [48,49]. Consequently, histone methylation is associated with either actively transcribed or repressed genes based on the specific methylated amino acid residue [50]. Although histone methylation does not directly cause transcriptional activation or repression, it modulates the transcriptional potential of genes [51]. The methylation marks are added to lysine or arginine residues in histone H3 or H4 by specific enzymes, i.e., histone lysine methyltransferases (HKMTs) or protein arginine methyltransferases (PRMTs) [52]. These methylation modifications can occur at different amino acids and involve various methylation states, including mono-, di-, or tri-methylation. Once established, the marks can be recognized by reader protein and removed by histone demethylases (HDMs) [52].

Different types of histone methylation have distinct biological functions [53,54]. In plants, the correlation between histone methylation and gene activation/repression depends on the specific methylation mark [55]. For instance, H3K4me3 and H3K36me3 are generally associated with active transcription, whereas genes marked with H3K27me3 or H3K9me3 often exhibit low transcript levels [56]. Significant differences exist between plants and animals. In animals, all forms of H3K4me modifications are associated with gene activation, whereas in plants, only H3K4me3 correlates with active transcription [57,58]. Regarding H3K9, about 40% of Arabidopsis coding genes are marked by H3K9me3, and only a minor fraction of the markers are found on TEs and pseudogenes [58,59]. Additionally, differences between plants and animals are also observed in H3K27 and H3K36 methylation patterns [60,61,62,63].

### 3.2. Global Alterations of Histone Methylation under Salt Stress

Recent studies have illustrated that changes in histone methylation are closely linked to activation or repression of gene expression [55,64,65]. The levels of H3K4me3 are increased while H3K27me3 levels are decreased, influencing downstream genes such as *RD29A/RD29B*, *AtHKT1*, and *RSM1* in response to high-salinity stress [64,66,67]. In Arabidopsis, some salt-related genes, such as *SUVH2/8* and *MSH6,* are downregulated with enhanced H3K9me2 under salt stress [68]. Moreover, in soybeans subjected to high-salinity stress, histone marks, including H3K4me2 and H3K4me3, are significantly upregulated, coordinating several key biological processes, such as stress response, cell wall modification, and ion homeostasis [69]. The upregulation of *Glyma20g30840*, *Glyma08g41450*, and *Glyma11g02400* genes under high-salinity stress is likely mediated by increased H3K4me3 and decreased H3K9me2 levels [67]. Furthermore, histone methylation changes at the *OsBZ8* gene locus are identified in both salt-tolerant and sensitive rice cultivars Nonabokra and IR64, respectively. Notably, Nonabokra rice obtains lower H3K27me3 and higher H3K4me3, while IR64 rice obtains higher H3K27me3 [70]. Additionally, in alfalfa, the transcription factor MsMYB4 plays a crucial role in salt stress response, with its activation correlated with increased levels of H3K4me3 and H3K9ac at specific promoter sites [71]. The changes in H3K4me3 and H3K9ac are primarily a result of the active gene expression rather than a cause, as *MsMYB4* expression was altered at 3 h after stress exposure [72].

### 3.3. Regulatory Mechanisms of Histone Methylation under Salt Stress

Salinity stress alters histone methylation levels through various regulatory pathways. In rice, the accumulation of AGO2 proteins at the *BIG GRAINS3* (*BG3*) locus leads to enhanced *BG3* expression. This is achieved by increasing H3K4me3 and decreasing H3K27me3 levels [73] (Figure 2). The histone demethylase JMJ15, which directly binds to and removes the H3K4me3 mark from the promoter and coding regions of *WRKY46* and *WRKY70*, mediates the repression of these *WRKY* genes, therefore contributing to increased salt tolerance in plants [66,74] (Figure 2). Plant homeodomain (PHD) finger proteins function as histone code readers that identify and attach to H3K4 marks on the H3 tail [75]. The soybean GmPHD6 specifically recognizes low levels of H3K4 methylation (H3K4me0/1/2) through its N-terminal domain but does not recognize H3K4me3. The GmPHD6 protein engages with its coactivator, LHP1-1/2, via its PHD finger to assemble a transcriptional activation complex. Overexpression of two GmPHD6 target genes, *CYP75B1* and *CYP82C4*, enhances stress tolerance of soybean [76] (Figure 2).

Under salt-stress conditions, SDG721, a SET DOMAIN GROUP protein possessing H3K4 methyltransferase activity, attaches to and adds the H3K4 mark within the promoter and coding regions of the *OsHKT1;5*, therefore regulating its expression levels [65,77]. During salt stress, removing H3K27me3 from *AtHKT1*, which is typically highly enriched, activates *AtHKT1* gene expression in Arabidopsis [46] (Figure 2). Additionally, a recent study indicates that salt stress induced the accumulation of H3 methylglyoxalation at the genomic loci of some salt-stress-responsive genes, which subsequently enhanced chromatin accessibility and gene expression [78]. Although several core enzymes, such as PHD6, JMJ15, and SDG721, have been identified, the regulatory mechanisms by which salt stress alters histone methylation remain unsolved.

### 3.4. Memorized Stress by Histone Methylation

DNA methylation imparts heritable stress memory to plants; however, this effect does not extend in the same manner as histone methylation [43,44,45,46,79,80] (Table 1). H3K4me3 plays a role in transcriptional memory, but in drought stress, this memory is short-lived and cannot be transmitted to the progenies [81]. Specifically, H3K4me3 is crucial for the activation of drought memory genes *GhP5CS1*, *GhNCED9*, *GhSnRK2*, and *GhPYL9-11A* during repeated drought stress. The levels of H3K4me3 associated with drought stress memory are diminished by the fifth day of the recovery period [42]. In contrast, the transcriptional memory triggered by heat stress often persists for up to a week. This response is typically associated with increased levels of H3K4 methylation [82]. Ding et al. found increased levels of H3K4me3 modification at trainable gene loci under repeated stress treatment, which was maintained during the recovery phase [83]. Another classic example is cold stress-induced memory of vernalization. Prolonged exposure to cold conditions leads to the repression of *FLC* expression, which is subsequently restored when the temperature rises in spring. Histone marks H3K4me3, and H3K36me3 positively regulate *FLC* expression, while H3K27me3 exerts an opposing effect on *FLC* expression and stress memory [36,84]. Additionally, a recent study has demonstrated that light modulates the salt-induced transcriptional memory through HY5-mediated regulation of H3K4me3 mark at the memory gene *P5CS1* [47].

## 4. Histone Acetylation

### 4.1. Histone Acetylation in Plants

The chromatin region with histone acetylation exhibits higher transcriptional activity. This is commonly attributed to the fact that the acetyl group neutralizes the positive charge of the histone, therefore altering the distance between nucleosomes and reducing the affinity between DNA and histone proteins [56,85]. Cells utilize two types of enzymes to modulate this dynamic process: histone acetyltransferases (HATs) and histone deacetylases (HDACs). HATs, which catalyze the acetylation of lysine residues, are further classified into four families: GNAT (GCN5-related N-acetyltransferase), MYST (Moz, Ybf2, Sas2, TIP60), p300/CBP, and TAFII250 [86]. Conversely, HDACs, which catalyze the removal of acetyl groups from lysine residues, are categorized into two classes: class I (including HDA 6/7/9/19) and class II (including 5/14/15/18) [87,88]. Despite some variations, HATs and HDACs exhibit conserved functions across a range of plant species, including *Arabidopsis thaliana*, *Oryza sativa*, foxtail millet [89], *Gossypium arboreum*, and kenaf [90]. These enzymes play essential roles in regulating various biological processes, such as ABA signaling transduction, SOS signaling transduction, ROS homeostasis maintenance, and LEA protein accumulation, all of which are critical for plant adaptation to salt stress [91,92,93]. Acetyl-coenzyme A (acetyl-CoA) is a pivotal metabolic intermediate that regulates essential cellular processes, including energy metabolism, mitosis, and autophagy. It functions as a crucial precursor for lipid synthesis and influences the acetylation profile of various proteins, notably histones [94,95]. The acetyl group from acetyl-CoA is transferred by histone acetyltransferases (HATs) to the ε-amino groups of lysine residues located at the N-terminal ends of histones. The acetyl-CoA utilized for histone acetylation is primarily generated by ATP-citrate lyase (ACL) in the tricarboxylic acid (TCA) cycle, occurring in either the mitochondria or the nucleus [96]. Consistently, mutations in the Arabidopsis *ATP-citrate lyase subunit A* (*ACLA*) result in decreased acetylation at H3K27 [97]. Nevertheless, direct evidence elucidating the acetyl-CoA promoted histone acetylation in response to salt stress remains to be unrevealed.

### 4.2. Global Alteration of Histone Acetylation under Salt Stress

Histone acetylation predominantly occurs at the K9, K14, and K27 residues of histone H3 and the K16 residue of histone H4. There is some acetylation in the plant salt regulatory gene region, and the acetylation pattern of the plant salt regulatory gene region will change greatly or slightly under salt stress, which will affect the plant’s tolerance to salt [70]. Recent studies have identified significant changes in the deacetylation of H3K9 and H3K14 under salt stress, resulting in the suppression of genes within the affected genomic regions. Notably, stress-induced acetylation of histone H3 is relatively rare compared to the deacetylation and subsequent gene repression observed under salt stress [98].

### 4.3. Regulatory Roles and Mechanisms of Histone Acetylation under Salt Stress

Histone acetylation plays a crucial role in regulating plant responses to salt stress by modulating various signaling pathways. Both histone acetylation and deacetylation serve as core regulators for ABA signaling pathways. For instance, Arabidopsis histone deacetylase HDA15 enhances deacetylation at the genomic region of *NCED3*, a gene involved in ABA biosynthesis. This process inhibits the binding of negative regulators to this genomic locus, ultimately promoting *NCED3* expression and ABA synthesis [99]. Conversely, another histone deacetylase, HDA710, which is induced by high salt stress and phytohormones such as jasmonic acid (JA) and abscisic acid (ABA), catalyzes the deacetylation of histones H3 and H4, therefore negatively regulating ABA signaling [100]. Furthermore, a poplar RPD3/HDA1-type histone deacetylase, 84KHDA909, has been shown to increase ABA accumulation in plants and alter the transcript abundance of ABA response genes when transferred to Arabidopsis [101]. In addition to its role in ABA signaling, histone acetylation and deacetylation also influence the SOS signaling pathway. Acetylation of histone H4 is essential for the activation of the *SOS1* gene [102]. Meanwhile, the transcription factor IDS1 (INDETERMINATE SPIKELET1), which belongs to the apetala2/ethylene response factor family, can collaborate with histone deacetylase HDA1 to repress *SOS1* (*SALT OVERLY SENSITIVE1*) expression by modulating H3 histone acetylation [93] (Table 2 and Figure 3).

In addition to their roles in signaling pathways, histone acetylation/deacetylation also regulates metabolites associated with salt stress. Plants utilize histone acetylation and deacetylation to manage reactive oxygen species (ROS). For instance, in wheat, TaHAG1 directly targets a subset of genes involved in hydrogen peroxide production, leading to H3 acetylation at these gene regions and, therefore, maintaining ROS homeostasis [108,112]. Similarly, in rice, analogous regulations have been observed [106]. Furthermore, the expression levels of beet *POX* genes, which are involved in ROS removal, are positively correlated with the levels of H3K9ac and H3K27ac [103]. In addition to regulating ROS removal, plants also modulate the synthesis of certain stress-responsive substances through histone acetylation and deacetylation to mitigate the effects of salt stress. Late embryogenesis abundant proteins (LEAs) are stress-induced proteins that enhance plant salt tolerance. The coordination of IDS1 and HDA1 negatively regulates LEA1 synthesis [93]. Similarly, another class I histone deacetylase, HDA19, exerts a comparable regulatory effect [87] (Table 2).

Under high-salinity conditions, root growth is inhibited, and root cells tend to swell, a process closely related to cell wall enlargement. In corn (*Zea mays*), the upregulation of the *ZmEXPB2* and *ZmXET1* genes, which are involved in cell wall modification under salt stress, is associated with increased H3K9 acetylation [105] (Figure 3). Wang et al. demonstrated that in *Chrysanthemum morifolium*, the heat shock factor A4 (CmHSFA4) recruits the corepressor TOPLESS (CmTPL) to inhibit the transcription of *CmMYB121*, a gene responsive to salt stress. This inhibition occurs through the reduction of H3 and H4 histone acetylation levels at the *CmMYB121* locus [111]. Additionally, in rice, the circadian clock regulatory core component OsPRR73 interacts with the histone deacetylase HDA10 to suppress the transcription of the Na^+^ absorption transporter *OsHKT2;1* in response to salt stress [107] (Table 2). Rice stomatal closure under salt stress is regulated by HDA704-mediated histone deacetylation of the *DST* and *ABIL2* genes [110] (Figure 3). Altogether, the plant utilizes histone acetylation and deacetylation as “on” and “off” switches to regulate gene expression levels, therefore modulating various pathways in response to the salt stress.

Histone acetylation does not function in isolation but often coordinates with other DNA and histone modifications. For example, under salt stress, both tobacco and Arabidopsis cells exhibit rapid upregulation of histone H3 Ser-10 phosphorylation, which is followed by subsequent phosphorylation of H3 and acetylation of histone H4 [109] (Table 2). Additionally, activation of four DNA-methylated-controlled transcription factors has been found to correlate with increased levels of histone H3K4 trimethylation and H3K9 acetylation [67].

## 5. Histone Variants

### 5.1. Histone Variants in Plants

Histone variants are non-allelic protein isomers, such as H2A.X, H2A.Z, macroH2A, CENP-A, and H3.3, that play crucial roles in chromatin structural diversification and gene expression. While they share sequence homology and major structural similarity with core histones, histone variants possess unique distributions and functions. The differential expression of histone variants at specific tissue and developmental stages indicates their specialized roles in modifying the structural and functional properties of chromatin [113]. Among eukaryotes, H2A is the most diverse histone, with its variants performing specialized functions during nucleosome assembly and genome packaging [114]. Consequently, histone variants of H2A are more prevalent across organisms, whereas other histone variants exhibit less diversity. Notably, the H2A family shows the greatest sequence differentiation at their C-terminus.

### 5.2. Regulatory Roles and Mechanisms of Histone Variants under Salt Stress

H2A.Z is a highly conserved histone variant that plays a critical role in regulating plant growth and development. A recent study has shown that H2A.Z is essential for salt tolerance in *Arabidopsis thaliana*. Under salt stress, H2A.Z is deposited at the promoter region near transcriptional initiation site (TSS) sites, influencing transcriptional regulation. However, the accumulation of H2A.Z is often negatively correlated with gene expression under salt stress [115]. Notably, in rice, the deposition of H2A.Z at the TSS is modulated by several stress-responsive regulators. For instance, in *osarp6* knockdown plants, the expression levels of stress-responsive genes, *ABA INSENSITIVE 1* (*ABI1*) and *ABA INSENSITIVE 2* (*ABI2*) are decreased [116]. The expression of the Arabidopsis transcription factor AtMYB44, which responds to salt stress, is regulated by H2A.Z deposition. Under salt stress, there is a marked reduction in H2A.Z deposition at the promoter of *AtMYB44*, which correlates with decreased occupancy of AtMYB44 in the same region [117]. This observation aligns with the general trend that gene expression levels are negatively correlated with H2A.Z enrichment under salt stress [118]. The rice H3 variant RH3.2A, which encodes the H3.2-type histone protein, shows upregulated expression in rice roots under both salt stress and ABA treatment [119].

Histone variants utilize specialized histone deposition mechanisms to ensure timely and site-specific binding to chromatin [120]. The regulation is partly mediated through the influence of histone modifications on nucleosomes and nucleosomes with specific deposition at relevant sites. ATP-dependent chromatin remodeling complexes modify nucleosome structure, therefore influencing the accessibility of packaged DNA sequences to trans-acting factors[121]. Histone variants further affect nucleosome dynamics following their deposition in conjunction with covalent post-translational modifications (PTMs) [119,122]. The precise mechanism by which histone variants mediate plant salt tolerance remains to be fully elucidated. However, transcriptomic analysis of histone variants in plants has provided valuable insights. For instance, Wang et al. conducted genome-wide characterization, phylogeny, and expression analysis of the histone gene family in cucumber [123].

## 6. Non-Coding RNAs

### 6.1. Non-Coding RNAs in Plants

MicroRNAs (miRNAs) are small, non-coding single-stranded RNAs, typically 21–24 nucleotides in length, that function as a gene regulator by modulating the abundance of their target genes. Plant miRNAs exhibit high complementarity to the specific sites of their target mRNAs, leading to the cleavage of most targeted mRNAs [124]. These miRNAs frequently target transcription factors involved in plant growth and development, therefore playing a crucial role in plant responses to abiotic stresses by regulating key transcription factors. Long non-coding RNAs (lncRNAs), which are non-coding RNAs longer than 200 nucleotides, are also vital in numerous biological processes, including dosage compensation, epigenetic regulation, cell cycle regulation, and cell differentiation. Both miRNAs and lncRNAs are significant regulatory elements in plants under salt stress, with many of these molecules showing altered expression in response to salt stress treatments.

### 6.2. Global Alterations of Non-Coding RNAs under Salt Stress

Under salt stress, the expression levels of miRNAs are significantly altered in a species-specific manner. In some plant species, the numbers of upregulated and downregulated miRNAs are comparable. For instance, in wheat, 49 miRNAs exhibited notable changes in expression levels under salt stress, with 25 showing significant upregulation and 24 showing significant downregulation [125]. A similar pattern was observed in the salt-sensitive broad bean Hassawi-3 [126]. Moreover, in contrast, most miRNAs are downregulated after salt stress treatment in some plant species. For example, in grapevine, 39 miRNAs were differentially expressed after salt stress, with 14 significantly upregulated and 25 significantly downregulated [127]. Comparable trends have been observed in the citrus root [128], rice [129], and the salt-tolerant cultivar *Fraxinus velutina* R7 [130]. Furthermore, in certain plant species, miRNA changes induced by salt stress are predominantly upregulated. In cotton, 51 miRNAs were upregulated, and 37 miRNAs were downregulated after 4 h of salt stress, while 48 miRNAs were significantly upregulated, and 27 miRNAs were downregulated after 5 days of long-term salt stress [131]. This pattern is similar to that observed in the *Fraxinus velutina* salt-sensitive cultivar S4 [130].

The salt-induced alterations of miRNA expression are time-dependent. For example, in fennel, six miRNAs were differentially expressed under salt stress; five were upregulated at 24 h, while one was downregulated. However, at 72 h post-salt stress, all studied miRNAs were upregulated [132]. A similar temporal pattern has been observed in members of the miR399 family in grapes [133]. Additionally, miRNA expression exhibits tissue specificity. A recent study reported the miRNA expression pattern in salt-tolerant Doc Phung (DP) rice under salt-stress conditions. Among 69 differentially expressed miRNAs, 50 miRNAs (five upregulated and 45 downregulated) were differentially expressed in shoot, while 28 miRNAs (13 upregulated and 15 downregulated) were differentially expressed in root tissue of the DP rice, respectively [129]. A similar pattern has been observed in carrot [134]. Notably, under salt stress, the expression pattern of a specific miRNA can vary across different plant species. For instance, miR156 is upregulated in *Arabidopsis thaliana*, *Raphanus sativus*, *Saccharum* spp., and *Suaeda maritima* but downregulated in *Populus trichocarpa* [124]. Similarly, miR159 is downregulated in *Arabidopsis thaliana*, *Nicotiana tabacum*, and *Oryza sativa* after salt stress, whereas it is upregulated in *Panicum virgatum*, *Saccharum* spp., and *Suaeda maritima* [135].

To regulate plant response to salt stress, the expression of long non-coding RNAs (lncRNAs) undergoes significant alteration. A study on tobacco has identified 2428 differentially expressed lncRNAs (DE-lncRNAs) in response to high-salinity treatment over time, with 2147 DE-lncRNAs detected in the roots and 495 in the leaves [136]. Functional predictions suggest that these DE-lncRNAs are involved in starch and sucrose metabolism pathways in roots and cysteine and methionine metabolism pathways in leaves. Additionally, under salt stress, 8724 lncRNA candidates were identified in the salt-tolerant rice species FL478, and 9235 lncRNA candidates were identified in the rice species IR19 [136]. In tomatoes, 154 and 137 lncRNAs exhibited differential expression in the M82 and *S. pennellii* varieties, respectively [137]. Functional analysis of target genes of these DE-lncRNAs in tomato indicates that some genes contribute to the salt-stress response by modulating the abscisic acid (ABA) signaling pathway [138].

### 6.3. Regulation of Core Stress-Responsive Genes by Non-Coding RNAs

Several studies have demonstrated that miRNAs play a crucial role in plant responses to salt stress by regulating hormone response pathways [139,140,141]. Notable examples include Arabidopsis miR165/166, grape miR390/394, tomato miR164, and Fraxinus miR393a/TIR1 [127,130,142]. The salt-induced downregulation of miR166 causes an upregulation of *PHB* expression, therefore triggering a salt-dependent rise in cytokinin levels through *IPT7* gene induction (Figure 4). Higher cytokinin levels at the transition zone activate the AHK3/ARR1/12 pathway, which promotes *SHY2* expression, which triggers cell differentiation and inhibits root meristem activity in response to salts [143]. Oxidative stress, a secondary consequence of plant salt stress, significantly impacts miRNA expression levels [144,145]. During salt stress, the accumulation of reactive oxygen species (ROS) inhibits the transcription of pre-miR169q, leading to an increased abundance of its target, *ZmNF-YA*8. The upregulation of *ZmNF-YA8*, in turn, promotes the expression of *ZmPERs*, which enhances peroxidase (POD) enzyme activity and contributes to the plant’s response to salt stress [146]. Additionally, differentially expressed miRNAs influence ROS accumulation in plants. For example, miR156a/b targets *SBP14* in *Fraxinus velutina*, downregulating *SBP14* expression under salt stress and therefore facilitating ROS clearance [130] (Figure 4). Furthermore, the knockdown of miR164a in tomatoes has been shown to reduce ROS accumulation in transgenic plants [142]. Conversely, the role of ROS as an upstream regulator of miR408 in maize remains speculative, with the precise relationship between them yet to be clarified [147]. Overexpression of *MIR408b* in maize has been shown to reduce lignin deposition, decrease the thickness of the pavement cell wall, and lower the number of cells in vascular bundles under salt stress. This suggests that miR408 may influence the influx of high Na^+^ concentration by regulating maize cell wall lignification, therefore impacting salt tolerance [147] (Table 3 and Figure 4). Glutathione (GSH), a tripeptide, plays a multifaceted role in plants’ responses to environmental stresses [148,149]. It functions as an antioxidant by mitigating oxidative stress, preventing lipid peroxidation, and protecting the plasma membrane. These actions subsequently reduce passive Na^+^ influx, therefore enhancing salt tolerance in plants. Furthermore, GSH is essential for maintaining cellular redox homeostasis under salt stress [148]. In the context of salt stress, the salt-insensitive tomato species *Lycopersicon pennellii* exhibits upregulation of both GSH biosynthesis and the activity of metabolizing enzymes compared to salt-sensitive tomato varieties [150]. Additionally, in salt-tolerant carrot varieties, increased expression of miR266 downregulates its target gene *gamma-glutamyl peptidase 1* (*GGP1*), resulting in elevated GSH levels and enhanced reactive oxygen species (ROS) scavenging efficiency [134]. Additionally, plants improve salt tolerance by regulating ion homeostasis, as demonstrated by the miR164d and the miR396a in *Fraxinus velutina* [130]. Notably, miR319 affects leaf phenotype and delays leaf senescence, therefore enhancing plant salt tolerance [151] (Figure 4).

Numerous lncRNAs have been identified as responsive to abiotic stress in plants, influencing ion transport and promoting signal transduction. Differentially expressed lncRNAs act as endogenous target mimics of certain miRNAs, such as rice miRNA osa-miR5809b, modulating their biological functions under salt stress [152]. In *Medicago truncatula*, the lncRNA MtCIR1 functions as a negative regulator of salt stress by enhancing abscisic acid (ABA) accumulation through the inhibition of the ABA catabolic enzyme *CYP707A2*, thus increasing seed germination sensitivity to salt stress. In *Tribulus terrestris* and *Arabidopsis thaliana*, MtCIR1 expression negatively regulates the salt-stress response by downregulating Na^+^ transporter genes, leading to higher Na^+^ accumulation in leaves and increased sensitivity of transgenic plants to salt-stress [153]. Differentially expressed poplar lncRNAs are observed between salt-tolerant and salt-sensitive cultivars [154] (Table 3). These lncRNAs may improve the adaptability of poplars to varying environmental conditions. Recent research has highlighted that poplar lncRNA.2-FL plays a crucial role in salt-stress tolerance in the FL478 cultivar by modulating 173 target genes *in trans* [155].

Both miRNA and lncRNA may play essential roles in plant stress memory. MiRNAs have been identified as key regulators of plant stress memory by mediating post-transcriptional silencing of target genes under stress conditions. Similarly, lncRNAs are involved in the formation of plant stress memories and contribute to enhanced stress tolerance [156]. Further research is needed to elucidate the unique regulatory mechanisms of miRNAs and lncRNAs in the formation of plant stress memory.

**Table 3 ijms-25-11698-t003:** Regulatory roles of non-coding RNAs in response to salt stress.

Non-Coding RNA	Species	Changes under Salt Stress	Target Genes and Biological Functions	References
MiR156	*Malus domestica* (apple)	Downregulation of *MIR156a*	Upregulation of *MdSPL13.* OE of MIR156a reduces salt tolerance	[157]
	*Zea mays*	Downregulation of *MIR156*	R2R3 Myb SBP-domain protein	[158]
MiR164	*Solanum lycopersicum* (tomato)	N.A.	KO of Sly-miR164a leads to reduced ROS and enhanced salt tolerance	[142]
	*Zea mays*	Downregulation of *MIR164*	*NAC1*, *ARF8*	[158]
MiR165/166	*Arabidopsis thaliana*	Downregulation of *MIR165A*, *MIR166A* and *MIR166B*	Salt stress induces *PHB* expression and production of cytokinin.	[143]
MiR168	*Oryza sativa*	N.A.	*PINHEAD* (*OsAGO1*). KD of miR168 leads to enhanced salt tolerance.	[159]
	*Zea mays*	Upregulation of *MIR168*	*AGO1*	[158]
MiR169	*Zea mays*	Downregulation of *zma-miR169* family members	*ZmNF-YA1*; *ZmNF-YA4*; *ZmNF-YA6*; *ZmNF-YA7*; *ZmNF-YA11*; *ZmNF-YA13*; *ZmNF-YA14*	[160]
MiR172	*Glycine max* (soybean)	Upregulation of gma-miR172a	*SSAC1*. OE of gma-miR172a leads to downregulation of *SSAC1* and enhanced salt tolerance.	[161]
	*Glycine max* (soybean)	Upregulation of miR172c	*NNC1*. OE of miR172c leads to enhanced salt tolerance.	[162]
	*Oryza sativa*	Upregulation of miR172a/b	*IDS1.* OE of miR172 leads to downregulation of *IDS1* and enhanced salt tolerance.	[163]
MiR319	*Arabidopsis thaliana*	Upregulation of miR319		[164]
	*Medicago truncatula* (model legume)	Downregulation of miR319	*TCP4*. OE of Mtr-miR319a leads to the downregulation of *TCP4* and enhanced salt tolerance.	[151]
	*Solanum linnaeanum* (eggplant)	Downregulation of miR319	TCP family transcription factor	[165]
	*Triticum aestivum*	Upregulation of miR319a		[166]
	*Zea mays*	Downregulation of miR319	*TCPs*	[158]
MiR390	*Populus* spp. (poplar)	Upregulation of miR390	*TAS3*. OE of miR390 leads to downregulation of *ARFs* (*ARF3.1*, *ARF3.2*, and *ARF4*) and enhanced salt tolerance.	[139]
MiR393	*Arabidopsis thaliana*	Upregulation of *MIR393A*	*TIR1*, *ABF2*, *ABF3*. Loss of miR393ab leads to an increase of lateral root number under salt stress, whereas OE of miR393 leads to enhanced salt tolerance.	[140,167]
	*Oryza sativa*	Upregulation of OsmiR393	*OsTIR1* and *OsAFB2*. OE of OsmiR393 leads to less tolerance to salt stress.	[168,169]
MiR394	*Arabidopsis thaliana*	Upregulation of miR394	*LCR*. OE of miR394 leads to less tolerance to salt stress.	[141]
MiR395	*Zea mays*	Upregulation of *MIR395*	NADP-dependent malic protein, ATP sulfurylase	[158]
MiR396	*Chrysanthemum indicum*	Upregulation of cin-miR396a	*CiGRF1* and *CiGRF5*. OE of cin-miR396a leads to less tolerance to salt stress.	[170]
	*Oryza sativa*	Upregulation of miR396b and downregulation of miR396c	*GRF6*. Loss of miR396 leads to enhanced salt tolerance.	[171,172]
	*Zea mays*	Upregulation of *MIR396*	Cytochrome oxidase	[158]
MiR397	*Arabidopsis thaliana*	Upregulation of miR397	*LAC2*, *LAC4*, and *LAC17*. OE of AtmiR397 leads to less tolerance to salt stress.	[173]
MiR399	*Arabidopsis thaliana*	Upregulation of miR399f	*CSP41b* and *ABF3*. OE of miR399f leads to enhanced salt tolerance.	[174]
MiR408	*Zea mays*	Downregulation of miR408	*ZmLAC9*. OE of miR408a/b leads to enhanced salt tolerance.	[147,175]
	*Salvia miltiorrhiza*	Upregulation of Sm-MIR408	OE of Sm-miR408 leads to enhanced salt tolerance.	[144]
MiR414	*Gossypium hirsutum* (cotton)	Downregulation of ghr-miR414c	*GhFSD1*. OE of ghr-miR414c leads to less tolerance to salt stress.	[145]
MiR528	*Oryza sativa*	Upregulation of miR528	*AO*. OE of miR528 leads to enhanced salt tolerance.	[176]
MiR1118	*Triticum aestivum*	Downregulation of miR1118	*PIP1;5.*	[177]
MiR1848	*Oryza sativa*	Upregulation of osa-miR1848	*OsCYP51G3*. OE of osa-miR1848 leads to less tolerance to salt stress.	[178]
Lnc_388, Lnc_883, Lnc_973, Lnc_253	*Gossypium hirsutum* (cotton)	Upregulation of Lnc_388, Lnc_883, Lnc_973, and Lnc_253	*LRR8* (Lnc_388), *msD3* (Lnc_883), miR399 (Lnc_973), and miR156 (Lnc_253). Loss of Lnc_973 leads to less tolerance to salt stress.	[179,180]
LncRNA354	*Gossypium hirsutum* (cotton)	Upregulation of LncRNA354	CeRNA for miR160b. Loss of LncRNA354 leads to enhanced salt tolerance.	[181]
Ptlinc-NAC72	*Populus trichocarpa*	Upregulation of Ptlinc-NAC72	*PtNAC72.A*/*B*. OE of Ptlinc-NAC72 leads to less tolerance to salt stress.	[182]
PUPPIES	*Arabidopsis thaliana*	Upregulation of PUPPIES	*DOG1*. Loss of PUPPIES leads to reduced expression of *DOG1*.	[183]
LncRNA77580	*Glycine max* (soybean)	N.A.	OE of LncRNA77580 leads to less tolerance to salt stress.	[184]
LncERF024	*Populus* ssp.	Upregulation of LncERF024	OE of LncERF024 leads to enhanced salt tolerance.	[154]
DRIR	*Arabidopsis thaliana*	Upregulation of DRIR	OE of DRIR leads to enhanced salt tolerance.	[185]

## 7. Conclusions and Perspectives

Since plants cannot proactively escape negative environments, they have evolved complex strategies to adapt to environmental stresses. Epigenetic regulation is a critical component of these adaptive mechanisms. This review collected and summarized recent studies on the overall state of DNA methylation, histone modifications, histone variants, and non-coding RNAs in plants, focusing on their global alterations in response to salt stress and their roles in enhancing salt tolerance (Figure 5). During salt stress, epigenetic regulators modulate genome architecture, reform the transcriptome, and establish a salt-specific regulatory network, therefore affecting physiological processes and plant phenotypes. Despite extensive research, several fundamental questions remain, such as how histone methyltransferases recognize their genomic targeting loci in response to salt stress, the mechanisms of DNA methyltransferase/demethylase recognition, and the deposition of H2A.Z. Most recent studies have focused on individual epigenetic processes, but further investigation is needed into the crosstalk and synergistic regulation between different epigenetic mechanisms in enhancing plant stress tolerance.

We also explore the emerging roles of epigenetic regulation in the establishment of plant stress memory (PSM), which is vital for plants to adapt to recurring stress. During stress, plants maintain a stress-responsive transcriptome. Post-stress, epigenetic marks may be inherited by subsequent generations through various mechanisms. However, the precise mechanisms and core factors involved in maintaining these epigenetic modifications to preserve genomic and transcriptional status remain unclear. Recent technological advancements, such as single-cell RNA sequencing (scRNA-seq), high-throughput chromosome conformation capture (Hi-C) [26], and CRISPR-Cas9 knockout tools [186], have advanced epigenetic research [186,187,188], potentially accelerating the systematic identification of PSM-associated factors. PSM and epigenetic engineering in crops could become crucial approaches for developing stress-resistant cultivars addressing severe global climate change [189]. A deeper understanding of the mechanisms underlying salt tolerance is essential for cultivating resilient crop varieties and ensuring food security. 

## Figures and Tables

**Figure 1 ijms-25-11698-f001:**
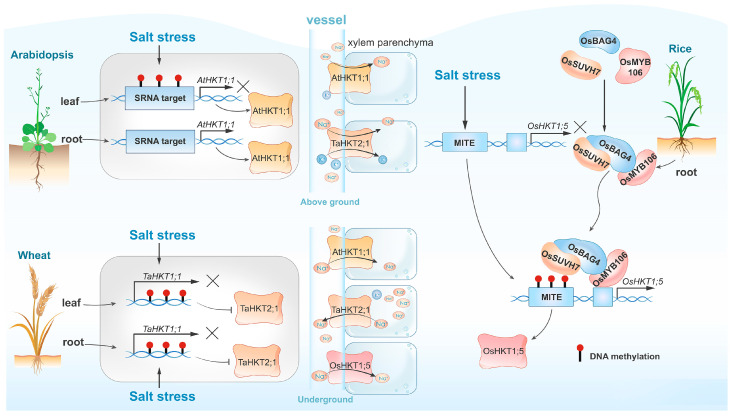
Comparison of DNA methylation-dependent *HKT* expression and salt tolerance across various plant species. In Arabidopsis (top left), salt-induced DNA methylation exhibits organ heterogeneity, resulting in the specific expression of *AtHKT1;1* in roots but not in leaves. In contrast, salt-induced DNA methylation suppresses the expression of the *TaHKT2;1* gene in both wheat leaves and roots (bottom left). The rice regulatory complex consisting of OsBAG4, OsMYB106, and OsSUVH7 recognizes salt-induced methylation in MITE sequences, therefore activating the expression of *OsHKT;5*. Variations in the expression levels of *HKT* genes influence Na^+^ transport within the vascular system, consequently impacting the plant’s salt tolerance.

**Figure 2 ijms-25-11698-f002:**
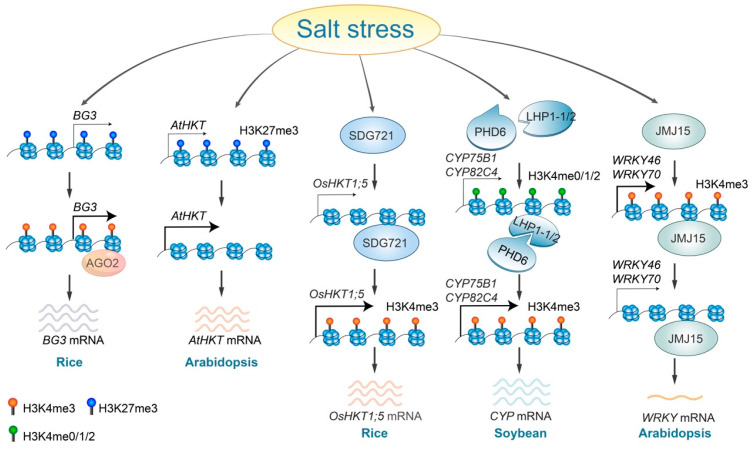
Regulatory roles and mechanisms of histone methylation in response to salt stress. Salt stress triggers an AGO2-dependent increase in H3K4me3 levels at the *BG3* locus, leading to the activation of *BG3* expression. In Arabidopsis, salt stress removes H3K27me3 at the *AtHKT* locus, promoting its expression, whereas in rice, the SDG721 protein is necessary to elevate H3K4me3 levels, therefore enhancing *OsHKT* expression. Additionally, salt stress facilitates the binding of the PHD6-LHP1-1/2 complex, resulting in increased H3K4me3 levels at the *CYP75B1* and *CYP82C4* loci and subsequently enhancing their expression. Conversely, JMJ15 removes H3K4me3 from the *WRKY46* and *WRKY70* loci in response to salt stress, therefore reducing gene expression levels.

**Figure 3 ijms-25-11698-f003:**
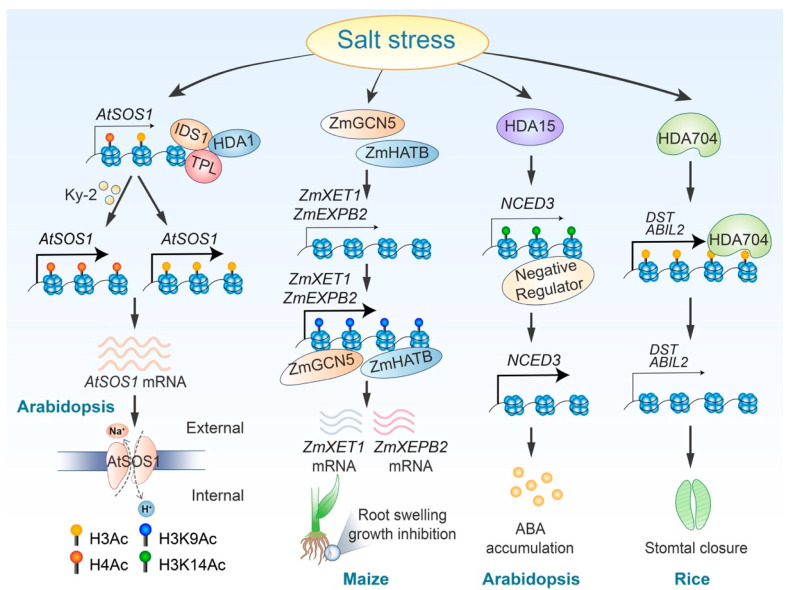
Regulatory roles and mechanisms of histone acetylation in response to salt stress. Ky-2 treatment inhibits the deacetylation of *AtSOS1*, therefore enhancing plant salt tolerance. Additionally, salt stress triggers the removal of the HDA1-IDS1-TPL complex, resulting in elevated H3 acetylation levels at the *AtSOS1* locus and increased *AtSOS1* expression. ZmGCN5 and ZmHATB promote H3K9 acetylation at the *ZmXET1* and *ZmEXPB2* loci. HDA15 reduces the binding of a negative regulator to the *NCED3* gene locus by removing H3K14 acetylation, consequently increasing *NCED3* expression. Furthermore, salt stress induces deacetylation of *DST* and *ABIL2* by HDA704, modulating stomatal closure.

**Figure 4 ijms-25-11698-f004:**
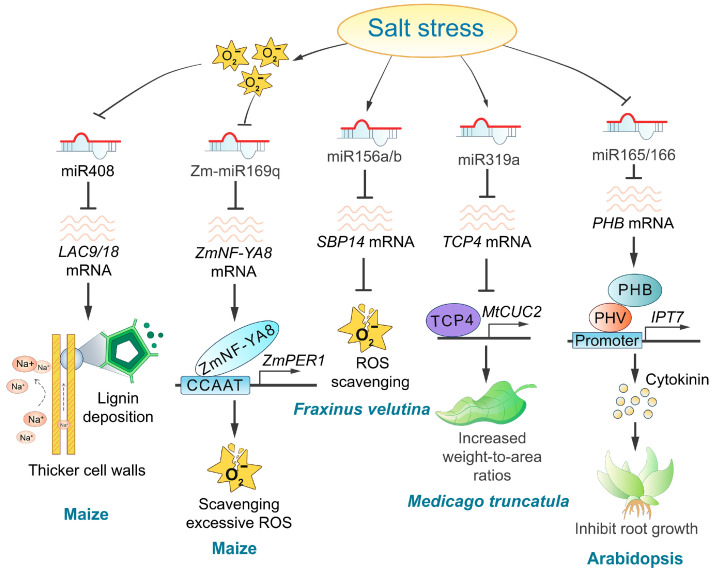
Regulatory roles and mechanisms of miRNAs in response to salt stress. Several miRNAs, including miR408, Zm-miR169q, and miR156a/b, play crucial regulatory roles in response to oxidative stress under high-salinity conditions. Additionally, other miRNAs, such as miR319a and miR165/166, are involved in the regulation of plant leaf senescence and phytohormonal responses to salt stress.

**Figure 5 ijms-25-11698-f005:**
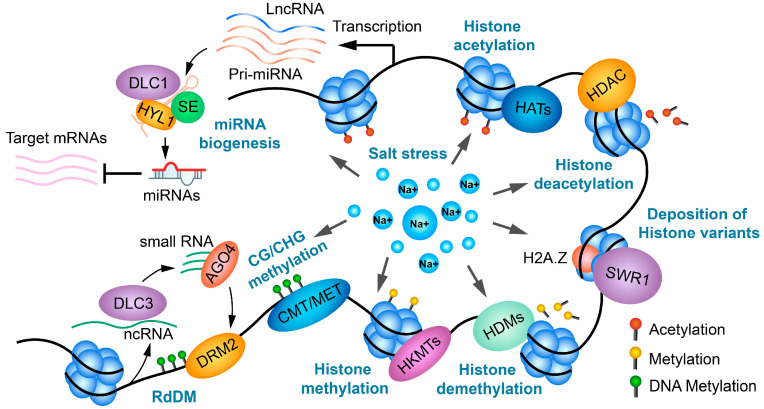
A schematic diagram illustrating all epigenetic factors and non-coding RNAs in response to salt stress.

**Table 1 ijms-25-11698-t001:** Memorized salt stress by DNA methylation and histone acetylation.

Epigenetic Modification	Species	Types	Description	References
DNA methylation	*Arabidopsis thaliana*	Long-term, transgenerational inheritance	Newly acquired stress tolerance and associated de novo DNA methylation are transmitted to the offspring. Progeny exposed to salt exhibited higher tolerance to stress. The PSM depends on altered DNA methylation and small RNA silencing pathways.	[43,44]
		Long-term, transgenerational inheritance	Salt-stress-altered DNA methylation was stably passed on to the next generation.	[37]
	*Thlaspi arvense*	Long-term, transgenerational inheritance	Salinity stress results in higher levels of epigenetic diversities, which are maintained in offspring, affecting the magnitude of phenotypic variation.	[45]
Histone methylation	*Arabidopsis thaliana*	Long-term, somatic memory	Salt treatment-induced shortening and fractionation of H3K27me3 islands affect somatic memory. For example, in primed plants, *HKT1* responded more effectively and rapidly to the second salt-stress event.	[46]
	*Arabidopsis thaliana*	Long-term, somatic memory	Light exposure is essential for salt-induced transcriptional memory to maintain H3K4me3 levels on the *P5CS1* gene.	[47]

**Table 2 ijms-25-11698-t002:** Regulatory roles of plant histone acetylation in response to salt stress.

Histone Acetylation Sites	Species	Target Genes	Changes under Salt Stress	References
H3K9	*Beta vulgaris*	*POX*	Acetylation	[103]
	*Glycine max*	*Glyma11g02400*, *Glyma08g41450*, *Glyma16g27950*, *Glyma20g30840*	Acetylation	[67]
	*Arabidopsis thaliana*	*DREB2A*, *RD29A*, *RD29B*	Acetylation	[104]
		*AtLIP4*, *AtLTP6*, *AtLIP3*, *AtPAD3*, *AtGST1*, *AtRAP2.6*, *AtMYB29*, *AtCYP79B2*, *AtGOLS2*, *AtPLC1*, *AtIMS3*, *AtANN1*, *AtAAP6*, *AtGSTF10*	Acetylation	[88]
		*AtANN4*	Deacetylation	
	*Zea mays*	*ZmEXPB2*, *ZmXET1*, *ZmHATB*, *ZmGCN*	Acetylation	[105]
	*Oryza sativa*	*OsBZ8*	Acetylation	[70]
		*OsMYB91*	Acetylation	[106]
		*OsHKT2;1*	Deacetylation	[107]
	*Triticum aestivum*	*TraesCS4D02G324800*, *TraesCS1D02G284900*, *TraesCS3D02G347900*	Acetylation	[108]
H3K14	*Arabidopsis thaliana*	*DREB2A*, *RD29A*, *RD29B*	Deacetylation	[104]
		*NCED3*	Deacetylation	[99]
	*Nicotiana tabacum*	*Tsi1*, *NtC7*	Acetylation	[109]
	*Triticum aestivum*	*TraesCS4D02G324800*, *TraesCS1D02G284900*, *TraesCS3D02G347900*	Acetylation	[108]
H3K27	*Beta vulgaris*	*POX*	Acetylation	[103]
	*Oryza sativa*	*OsBZ8*	Acetylation	[70]
H4K5	*Zea mays*	*ZmHATB*, *ZmGCN5*	Acetylation	[105]
H4K16	*Arabidopsis thaliana*	*NCED3*	Deacetylation	[99]
H3	*Oryza sativa*	*OsLEA3*, *OsABI5*, *OsbZIP72*, *OsNHX1*	Acetylation	[100]
		*LEA1*, *SOS1*	Acetylation	[93]
		*DST*, *ABIL2*	Deacetylation	[110]
	*Chrysanthemum morifolium*	*CmMYB121*	Acetylation	[111]
H4	*Arabidopsis thaliana*	*AtSOS1*	Acetylation	[102]
	*Oryza sativa*	*OsLEA3*, *OsABI5*, *OsbZIP72*, *OsNHX1*	Acetylation	[100]
		*DST*, *ABIL2*	Deacetylation	[110]
	*Chrysanthemum morifolium*	*CmMYB121*	Acetylation	[111]

## Data Availability

Not applicable.

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
