# Peer review of "Insights into the Epigenetic Basis of Plant Salt Tolerance"

_ijms, 2024, doi:10.3390/ijms252111698_

Round 1

Reviewer 1 Report

Comments and Suggestions for Authors

Dear Authors,

Great review. It touches nearly all levels of salinity tolerance (it would be interesting to read about heavy metals, too) and is a comprehensive study regarding salinity tolerance. Still, it is worth mentioning biochemical aspects of salinity. For sure, focusing on  Ac-CoA (the Krebs cycle), SAM (the Yang cycle), and GSH (transsulfuration pathway) in terms of signal transduction would improve the ms significantly. The other suggestion concerns figures. Adding descriptions (not only titles) would help read the review. Showing a general scheme (even somewhat speculative) linking all levels involved in salinity would assist a reader in finding a proper perspective of the discussed issues (figure plus description). Some minor comments are related to the text formatting, which the authors may quickly correct.

Hope to see it published soon.

Author Response

Comments 1: Great review. It touches nearly all levels of salinity tolerance (it would be interesting to read about heavy metals, too) and is a comprehensive study regarding salinity tolerance. Still, it is worth mentioning biochemical aspects of salinity.

Response1: Thank you for your careful evaluation and insightful suggestions for our manuscript. Your comments and suggestions help us to improve academic rigor of our manuscripts. Presented below is a detailed, point-by-point response addressing the reviewers’ comments and concerns. The revised text has been highlighted in red color within revised manuscript.

Comments 2: For sure, focusing on Ac-CoA (the Krebs cycle), SAM (the Yang cycle), and GSH (transsulfuration pathway) in terms of signal transduction would improve the ms significantly.

Response 2: We agree with this comment. We have added three paragraphs summarizing the responses to salt stress involving Ac-CoA (the Krebs cycle), SAM (the Yang cycle), and GSH (the transsulfuration pathway) biosynthesis, as well as their signaling transduction pathways. Additionally, we discuss their effects on the deposition of epigenetic modifications. The revised text has been highlighted in red color within revised manuscript.

Comments 3: Adding descriptions (not only titles) would help read the review. Showing a general scheme (even somewhat speculative) linking all levels involved in salinity would assist a reader in finding a proper perspective of the discussed issues (figure plus description).

Response 3: We agree with this comment. We have incorporated detailed descriptions for each figure to enhance reader comprehension. The revised text has been highlighted in red color within revised manuscript.

Comments 4: Some minor comments are related to the text formatting, which the authors may quickly correct.

Response 4: We agree with this comment. We have revised the text formatting for clarity.

Reviewer 2 Report

Comments and Suggestions for Authors

Here’s my evaluation for the review article ID: ijms-3221878 entitled as “Zhang et al., Insights into the epigenetic basis of plant salt tolerance.

The review article was articulated very well with pertinent tables and figures. Several and latest relevant literatures/references were reviewed with sound discussion, and conclusion.

Question: - Where is the figure 5 mentioned on page 13, line#449+, 467 & 470. I think this figure must be missed.  Or need clarification.

Thank you

Author Response

Comments 1: Here’s my evaluation for the review article ID: ijms-3221878 entitled as “Zhang et al., Insights into the epigenetic basis of plant salt tolerance.” The review article was articulated very well with pertinent tables and figures. Several and latest relevant literatures/references were reviewed with sound discussion, and conclusion.

Response 1: Thank you so much for your patience and helpful suggestions. Your comments and suggestions have significantly contributed to elevating the quality of our manuscript. Presented below is a detailed, point-by-point response addressing the reviewers’ comments and concerns. The revised text has been highlighted in red color within revised manuscript.

Comments 2: Where is the figure 5 mentioned on page 13, line#449+, 467 & 470. I think this figure must be missed.  Or need clarification.

Response 2: Thank you for pointing this out. This is an incorrect label for Figure 4, and we have made the necessary corrections in the revised manuscript. The revised text has been highlighted in red color within revised manuscript.

Reviewer 3 Report

Comments and Suggestions for Authors

The manuscript “Insights into the epigenetic basis of plant salt tolerance” discusses and summarizes the role of epigenetics factors (including DNA methylation, histone methylation, and acetylation) and non-coding RNA (including long non-coding RNA and miRNA) in response to salt stress. The inheritability of plant stress memory (changes in DNA methylation) in order to adapt the salt stress was included in this mini review to enhance the plant tolerance to salt stress and are important to agricultural breeding.

Minor errors

Line 525 to 528, no information was filled in this section including funding, please fill in the necessary information

Other suggestions

A schematic diagram depicting all the of epigenetic factors and non-coding RNAs in a network in response to salt stress can be included to have a better picture of this review

Since regulatory role and mechanism of non-coding RNA is quite distinct among different plant species, more obvious labeling in specific plant species is required in the figure, as demonstrated in Figure 1

Author Response

Comments 1: The manuscript “Insights into the epigenetic basis of plant salt tolerance” discusses and summarizes the role of epigenetics factors (including DNA methylation, histone methylation, and acetylation) and non-coding RNA (including long non-coding RNA and miRNA) in response to salt stress. The inheritability of plant stress memory (changes in DNA methylation) in order to adapt the salt stress was included in this mini review to enhance the plant tolerance to salt stress and are important to agricultural breeding.

Response 1: Thank you for your correspondence and the insightful feedback regarding our manuscript. These comments hold great value for revising and improving our manuscript. Presented below is a detailed, point-by-point response addressing the reviewers’ comments and concerns. The revised text has been highlighted in red color within revised manuscript.

Comments 2: Line 525 to 528, no information was filled in this section including funding, please fill in the necessary information.

Response 2: Thank you for pointing this out. This research was funded by National Natural Science Foundation of China under grant number 32170283. The funding information has been incorporated into the manuscript and is highlighted in red.

Comments 3: A schematic diagram depicting all the of epigenetic factors and non-coding RNAs in a network in response to salt stress can be included to have a better picture of this review.

Response 3: Thank you for pointing this out. We have added a new Figure, Figure 5, which presents a schematic diagram illustrating all epigenetic factors and non-coding RNAs in response to salt stress. Figure 5 has been included in the revised manuscript.

Comments 4: Since regulatory role and mechanism of non-coding RNA is quite distinct among different plant species, more obvious labeling in specific plant species is required in the figure, as demonstrated in Figure 1.

Response 4: Thank you for pointing this out. We have included the names of plant species in each figure to enhance clarity and facilitate understanding. New Figure 2, 3 and 4 has been added into the revised manuscript.